# Dissemination of OXA-48- and NDM-1-Producing Enterobacterales Isolates in an Algerian Hospital

**DOI:** 10.3390/antibiotics11060750

**Published:** 2022-05-31

**Authors:** Amel Abderrahim, Nassima Djahmi, Lotfi Loucif, Sabrina Nedjai, Widad Chelaghma, Djamila Gameci-Kirane, Mazouz Dekhil, Jean-Philippe Lavigne, Alix Pantel

**Affiliations:** 1Département de Biochimie, Faculté des Sciences, Université Badji Mokhtar Annaba, Annaba 23000, Algeria; amelabderrahim@yahoo.fr (A.A.); djamkirane@yahoo.fr (D.G.-K.); 2Laboratoire de Microbiologie, CHU Ibn Rochd, Annaba 23000, Algeria; djahmin@hotmail.fr (N.D.); nedjaisabrina@yahoo.fr (S.N.); mdekhil23@yahoo.fr (M.D.); 3Laboratoire de Biotechnologie des Molécules Bioactives et de la Physiopathologie Cellulaire (LBMBPC), Faculté des Sciences de la Nature et de la Vie, Université de Batna 2, Batna 05000, Algeria; lotfiloucif@hotmail.fr; 4Laboratoire de Microbiologie Appliquée à l’Agroalimentaire au Biomédical et à l’Environnement, Département de Biologie, Faculté des Sciences de la Nature et de la Vie et Sciences de la Terre et de l’Univers, Université Abou Bekr Belkaid, Tlemcen 13000, Algeria; widadc2014@hotmail.com; 5Virulence Bactérienne et Infections Chroniques, INSERM U1047, Université Montpellier, Service de Microbiologie et Hygiène Hospitalière, CHU Nîmes, 30900 Nîmes, France; alix.pantel@chu-nimes.fr

**Keywords:** Algeria, carbapenemases, community, Enterobacterales, NDM-1, OXA-48, University Hospital

## Abstract

Multidrug-resistant (MDR) Enterobacterales remain an increasing problem in Algeria, notably due to the emergence of carbapenemase producers. We investigated the molecular characteristics of carbapenem-resistant Enterobacterales isolates recovered from outpatients and inpatients in Eastern Algeria. Non-repetitive Enterobacterales with reduced susceptibility to carbapenems were consecutively collected from clinical specimens in Annaba University Hospital (Algeria) between April 2016 and December 2018. Isolates were characterized with regard to antibiotic resistance, resistome and virulome content, clonality, and plasmid support. Of the 168 isolates analyzed, 29 (17.3%) were carbapenemase producers and identified as *K. pneumoniae* (*n* = 23), *E. coli* (*n* = 5), and *E. cloacae* (*n* = 1). *bla*_OXA-48_ was the most prevalent carbapenemase-encoding gene (*n* = 26/29), followed by *bla*_NDM-1_ gene (*n* = 3/29). *K. pneumoniae* isolates harbored some virulence traits (*entB*, *ugeF*, *ureA*, *mrkD*, *fimH*), whereas *E. coli* had a commensal origin (E, A, and B1). Clonality analysis revealed clonal expansions of ST101 *K. pneumoniae* and ST758 *E. coli*. Plasmid analysis showed a large diversity of incompatibility groups, with a predominance of IncM (*n* = 26, 89.7%). A global dissemination of OXA-48-producing Enterobacterales in the Algerian hospital but also the detection of NDM-1-producing *E. coli* in community settings were observed. The importance of this diffusion must be absolutely investigated and controlled.

## 1. Introduction

Antimicrobial resistance has become a major threat for public health worldwide. Currently, the emergence of carbapenem resistance is gaining considerable attention because these drugs were considered for a long time as the last-line agents for the treatment of critical infections caused by multidrug-resistant Enterobacterales [1]. Their use has significantly increased since the 1980s, due to the rapid spread of extended-spectrum β-lactamases (ESBL) producers worldwide [2], which led to the emergence of carbapenem-resistant Enterobacterales. The first detected carbapenemase type, NmcA, was described in 1993 among *Enterobacter cloacae* isolates [3]. Since then, increasing studies on carbapenem resistance in Enterobacterales species have been reported in many countries, which are mainly caused by the spread of carbapenemase producers [4]. Three carbapenemase classes have been identified in Enterobacterales. These classes include class A serine enzymes (KPC), class B metallo-β-lactamases (VIM, IMP, NDM), and class D enzymes (OXA-48-like) [5]. The OXA-48 carbapenemase was first identified in *Klebsiella pneumoniae* in 2001 in Turkey. The OXA-48-like β-lactamase has spread in several regions, gradually becoming endemic in North Africa and Turkey while emerging in several other parts of the world, including Europe and other African countries [5]. The New Delhi metallo-β-lactamase (NDM) enzyme group is among the recent described MBLs, and is responsible for major health concerns due to its rapid propagation in all continents and its severe clinical impact [6].

Carbapenemase-producing Enterobacterales (CPE) strains are often multidrug- or pan-drug-resistant, leaving only very few therapeutic options for treating serious infections. In addition, the co-occurrence of carbapenemase with other resistant determinants such as aminoglycosides and quinolones resistance is a cause for global health concern. In Algeria, data concerning the emergence and diffusion of the CPE strains are available [5,7]. OXA-48-like enzymes have been extensively reported [5], notably, in different ecological niches (human, livestock, wild animals, and wildlife), and are thought to be endemic in Algeria. However, KPC- and NDM-producing Enterobacterales have also been observed in humans and the environment [8,9,10,11,12].

Thus, the aim of the present study was to describe the molecular epidemiology (resistance and virulence determinants, clonality) of Enterobacterales isolates with reduced susceptibility to carbapenems recovered from outpatients and inpatients in Annaba University Hospital, in Algeria.

## 2. Results

### 2.1. Phenotypic Characterization and Antibiotic Susceptibility

During the studied period, 168 non-repetitive Enterobacterales isolates with reduced susceptibility to carbapenems were included in our study. Using a KPC, MBL, and OXA-48 Confirm kit [12], 29 of these 168 isolates (17.3%) were suspected to produce a carbapenemase, with 26 belonging to class D and 3 to class B. The other isolates (*n* = 139) were non-carbapenemase-producing Enterobacterales with an impermeability of the outer membrane. The bacterial isolates were identified as *K. pneumoniae* (*n* = 23), *Escherichia coli* (*n* = 5), and *E. cloacae* (*n* = 1). All the isolates were resistant to ertapenem, 86.2% (*n* = 25) to imipenem, and 69.0% (*n* = 20) to meropenem. Variable rates of resistance were noted for the other antibiotics, with 82.8% (*n* = 24) to fluoroquinolones (ofloxacin and ciprofloxacin) and cefoxitin, 79.3% (*n* = 23) to cefotaxime, ceftazidime, and tobramycin, 75.9% (*n* = 22) to cefepime, 72.4% (*n* = 21) to aztreonam and gentamicin, and 37.9% (*n* = 11) to amikacin. All isolates were susceptible to colistin (Appendix A).

### 2.2. Molecular Detection of Resistant Determinants

Using molecular tools, we definitively confirmed 29 carbapanemase producers, of which 26 isolates (89.7% of the carbapenemase producers) harbored the *bla*_OXA-48_ gene, including twenty-one *K. pneumoniae*, four *E. coli*, and one *E. cloacae*. The *bla*_NDM-1_ gene was detected in the three remaining isolates, with two *K. pneumoniae* isolates and one *E. coli* isolate (Table 1). 

In addition, 22 CPE isolates (75.9%) also contained a *bla*_CTX-M_ gene, of which 21 (95.5%) *bla*_CTX-M-15_ and 1 (4.5%) *bla*_CTX-M-9_.

Interestingly, these isolates co-harbored different PMQR and aminoglycoside-modifying enzyme-encoding genes: *oqxAB* (*n* = 23, 79.3%), *aac6*′-Ib-cr (*n* = 22, 75.9%), *aac3*-II (*n* = 19, 65.5%), *qnrB* (*n* = 15, 51.7%), *qnrS* (*n* = 12, 41.4%), *qnrA* (*n* = 10, 34.5%), and *ant2*″ (*n* = 1, 3.4%). The methylase-encoding gene *rmtB* was detected in two isolates (6.8%): one NDM-1-producing *E. coli* and one OXA-48-producing *K. pneumoniae* isolate (Table 1). Finally, we noted association between *qnrB* and *aac(6′)*-Ib-cr in twelve isolates (41.4%) and *qnrB* and *aac(6′)*-Ib-cr in the two NDM-1-producing *K. pneumoniae*, and in twelve OXA-48-producing isolates (57.1%).

### 2.3. Virulence Profiles of the Carbapanemase-Producing Enterobacterales

The evaluation of the virulence of the carbapenemase-producing *K. pneumoniae* isolates by specific PCR detection of the main virulence factors (VF) genes showed that no hypervirulent strain has been identified. However, the distribution of the 18 virulence-associated genes indicated that the isolates harbored between six and eight genes per isolate (Table 1). All isolates harbored *entB*, *ugeF*, *ureA*, *mrkD*, and *fimH* genes. *ybtS* gene was detected in 95.7% of isolates (*n* = 22), and *wabG* in 91.3% of isolates (*n* = 21). *kfu* was observed in 73.9% of the isolates (*n* = 17). *iutA* was only detected in one isolate. Neither capsular serotype genes nor hypermucoviscous phenotype were noted (Table 1).

The *E. coli* isolates belonged to the phylogroup E (*n* = 3/5), the B1 phylogroup (*n* = 1), or the A phylotype (*n* = 1). These isolates presented a low content of VF genes with only *traT* and *fimH* detected in all the isolates (*n* = 5), *hlyA* in three, *iutA* in two, and the other genes in one isolate (*irp2*, *malX*, *papAH*, *papC*, *papG2*, and *kpsMT2*), suggesting that *E. coli* had a commensal origin, except the Z27 isolate, which could be considered as an Extraintestinal pathogenic *E. coli* (ExPEC).

### 2.4. Clonal Relationships among Species

MLST analysis revealed that ST101 was the most common ST (*n* = 16, 69.6%) for *K. pneumoniae*. All these isolates harbored the *bla*_OXA-48_ gene. The other ST types identified were ST147 (*n* = 4), ST48 (*n* = 1), ST113 (*n* = 1), and ST405 (*n* = 1). Among the four ST147, two *K. pneumoniae* isolates carried the *bla*_NDM-1_ gene. The Rep-PCR results showed a large diversity of the isolates. Among *K. pneumoniae*, 11 clusters were observed (Figure 1), with a main cluster (pattern V) including 11 isolates (47.8%): Z6, Z11, Z12, Z13, Z14, Z16, Z21, Z23, Z27, Z33, and 859.

The four OXA-48-positive *E. coli* strains were totally diverse, belonging to ST44, 758, 759, and 834 (Figure 2). The last *E. coli* isolate (56) harboring *bla*_NDM-1_ belonged to ST759 and was clonal to the 58B isolate.

Finally, the OXA-48-producing *E. cloacae* belonged to ST68.

### 2.5. Plasmid Typing of the Carbapanemase-Producing Enterobacterales

The PBRT showed a large diversity of plasmids identified as follows: IncL, Mobc11, IncR, MobQU, colE NT2, FIIK, FV, IncAC, X3, colE NT1, and colE NT2 plasmid type. The different plasmids identified in the CPE isolates are summarized in Table 1.

IncM was the most prevalent (*n* = 26/29, 89.7%), notably, in OXA-48 producers (*n* = 26/26 isolates, 100%). IncFV was the most frequent plasmid detected in *E. coli* (*n* = 5/6 isolates, 83.3%). Finally, IncR and IncFV groups (*n* = 2 and *n* = 1, respectively) were detected in NDM-1-producers.

## 3. Discussion

The emergence and spread of carbapenemase-producing Gram-negative bacteria has become a major public health problem worldwide including North African countries. In developing countries, such as Algeria, several studies on carbapenemase producers have been reported [13,14,15,16,17,18,19]. This study demonstrated that OXA-48 enzyme is the most prevalent carbapenemase among patients admitted in our hospital, supporting the previous finding that OXA-48 is endemic in Algeria and is commonly disseminated in the Mediterranean region [5]. These results are in accordance with various reports from Algeria reporting the isolation of OXA-48-producing enterobacterial isolates, specifically, *K. pneumoniae*, *E. coli*, and *E. cloacae* from various ecological niches [5,20,21]. In this investigation, we also reported the isolation of NDM-1-producing Enterobacterales isolates. The first identification of a *bla*_NDM_ gene in a clinical strain was described in 2009 in a Swedish patient of Indian origin, hospitalized in Sweden [22]. Following this report, NDM producers have been shown to be globally distributed, with virtually all countries, where the Asian continent, particularly India, the Middle East, and Balkan states serve as a reservoir of *bla*_NDM_ genes [6,23]. In Algeria, the *bla*_NDM-1_ gene was detected mostly in *A. baumannii* isolates [19]. However, the first case of NDM-1-producing *K. pneumoniae* was identified in Annaba University Hospital from a patient with a urinary tract infection in 2017 [14]. Initially, CPE appeared to cause hospital-acquired infections, but more recently it has spread to the community setting [17,24]. To the best of our knowledge, here we report for the first time the detection of *bla*_NDM-1_ gene in *E. coli* isolate causing community-acquired infection in Algeria. This later co-harbored also the *bla*_CTX-M-1_, *aac6′-Ib-cr*, and *rmtB* genes. In Algeria, the first detection of CPE from a community setting through the isolation of OXA-48-producing *K. pneumoniae* isolate in community-acquired urinary tract infection was described in Batna city [24]. Since then, only one study has described the isolation of OXA-48-producing *E. coli* and *K. pneumoniae* in Annaba University Hospital [17]. Different carbapenemase producers were described from community-acquired infection worldwide. In India, *bla*_NDM-5_ has previously been reported in *E. coli* strains co-harboring extended-spectrum β-lactamases genes in community-acquired urinary tract infection [25]. In addition, *bla*_NDM-1_, *bla*_NDM-5_, and *bla*_OXA-181_ genes were also reported in *E. coli* isolates co-harboring the *bla*_CTX-M-15_ ESBL gene in Riyadh, Saudi Arabia, and in Czech Republic [26,27]. CPE commonly exhibit multidrug-resistant or extensively drug-resistant phenotypes, limiting treatment options. This was also observed in our strain collection due to the association of a wide variety of other acquired resistance genes, including those conferring resistance, with aminoglycosides and quinolones.

MLST analysis of *K. pneumoniae*, *E. coli*, and *E. cloacae* isolates showed the occurrence of several clones, with the ST101 in *K. pneumoniae* isolates being the most frequent. Indeed, in Algeria, the international high-risk resistant lineages ST101 carrying *K. pneumoniae* harboring the *bla*_OXA-48_ gene had accounted for an outbreak in Batna University Hospital, Algeria [4]. Interestingly, this epidemic clone has been involved in several outbreaks in Spain [28,29]. Therefore, ST101 was reported as a predominant OXA-48-producing *K. pneumoniae* clone in various Mediterranean countries. In addition, the ST101 is also reported as multidrug-resistant clones present in several African, American, and European countries [30]. ST147, the second most common clone in this study, has been found among three OXA-48-producing strains and two isolates harboring *bla*_NDM-1_ genes. This other international high-risk resistant lineage has recently been reported to be the main clone of *K. pneumoniae* strains carrying the *bla*_NDM-1_ gene in Annaba University Hospital [2]. Another report in the same city isolated OXA-48-producing *K. pneumoniae* ST147 isolates in 2018 [17]. These two sequence types, ST101 and ST147, are known to have strong epidemic potential and have been associated with various carbapenemases such as NDM-1 with ST147 in Yemen [31], VIM with ST147 in Greece [32], and ST101 with OXA-48 in Tunisia and Germany [33,34]. Furthermore, ST405 was associated in this work with one *K. pneumoniae* strain co-harboring the *bla*_OXA-48_, *bla*_CTX-M-15_, *aac6′*-1b-cr, *aac3-II*, *oqxAB*, and *qnrB* genes. The sequence type ST405 is also described as a multidrug-resistant clone present in Algeria from inanimate surfaces in hospital environments [35], from ready-to-eat sandwiches [36], and also in various European countries including Italy [37] and Spain [30]. Finally, our study reports for the first time the detection of OXA-48-producing *E. cloacae* ST68 as well as the first *K. pneumoniae* ST113, and *E. coli* ST758, ST44, and ST834 in Algeria. ST44 has been reported in household cockroaches in Ghana [38], in Ecuador from fresh vegetables samples [39], and in Chile from migratory Franklin’s Gulls (*Leucophaeus pipixcan*) [40]. In our study, the association of multidrug-resistant phenotypes with the detection of multiple VFs notably in *K. pneumoniae* isolates poses a serious global health risk, which increases the mortality rates. The diversity of the virulence genes within clones from the same hospital suggests that they were mostly acquired horizontally rather than vertically and could be associated with mobile genetic elements integrated into the chromosome. Further investigations must be performed to corroborate this result.

## 4. Materials and Methods

### 4.1. Bacterial Collection

From April 2016 to December 2018, all non-repetitive Enterobacterales isolates with reduced susceptibility to carbapenems (intermediately or resistant to at least one carbapenem molecule) were consecutively collected from different clinical specimens of both inpatients and outpatients in Annaba University Hospital (Algeria). The isolates were identified using the Vitek^®^ MS system (bioMérieux, Marcy l’Etoile, France).

### 4.2. Antibiotic Susceptibility Testing

Susceptibility to antimicrobial agents was tested by the disk diffusion method (Bio-Rad, Marnes La Coquette, France) on Mueller–Hinton agar according to EUCAST recommendations (www.eucast.org, accessed on 20 January 2022). The following antibiotics were tested: amoxicillin (20 μg), cefoxitin (30 μg), cefotaxime (30 μg), ceftazidime (30 μg), cefepime (30 μg), aztreonam (30 μg), amoxicillin–clavulanic acid (20–10 μg), ertapenem (10 μg), meropenem (10 μg), imipenem (10 μg), gentamicin (10 μg), amikacin (30 μg), tobramycin (10 μg), ofloxacin (5 μg), and ciprofloxacin (5 μg). The minimal inhibition concentrations (MIC) of colistin and carbapenems (ertapenem, imipenem, and meropenem) were measured, respectively, by broth microdilution (UMIC^®^, Biocentric, Bandol, France) and E-test strips (bioMérieux), and interpreted following the EUCAST Breakpoints (v1.0, 2021). The non-susceptibility to the carbapenems and colistin was assessed on the MICs cut-offs: ertapenem: >0.5 mg/L, imipenem and meropenem: >2 mg/L, and colistin: >2 mg/L.

### 4.3. Phenotypic Characterization of Carbapenemase Production

Detection of carbapenemase producers was investigated using the KPC, MBL, and OXA-48 Confirm kit (Rosco Diagnostica A/S, Taastrup, Denmark) following the manufacturer’s recommendations [12].

### 4.4. Molecular Characterization of Antibiotic Resistance Genes

Total DNA was extracted using the EZ1 DNA Tissue kit on the BioRobot EZ1 extraction platform (Qiagen, Courtaboeuf, France). Carbapenemase-encoding genes (*bla*_KPC_, *bla*_VIM_, *bla*_IMP_, *bla*_NDM-1_, *bla*_OXA-48-like_) and extended-spectrum β-lactamases (ESBLs) determinants (*bla*_TEM_, *bla*_SHV_, and *bla*_CTX-M_) were screened by multiplex PCR using specific primers and confirmed by sequencing, as previously described [41,42]. Simplex or multiplex PCR for detecting aminoglycoside-modifying enzymes (*aac(3)*-I, *aac(3)-II*, *aac(3)*-IV, *aac(6′)*-Ib, *ant(2″)*, *aph(3′)*-Ia, *aph(3′)*-VI), and the 16S rRNA methyltranferases genes (*armA*, *rmtA*, *rmtB*, *rmtC*, *rmtD*, *rmtE*, *rmtF*, *npmA)* was carried out [43,44,45]. In addition, plasmid-mediated quinolone resistance genes (*qnrA*, *qnrB*, *qnrS*, *oqxA*, *oqxB*, *qepA*, and *aac(6′)*-Ib/Ib-*cr*) were also investigated by multiplex PCR as described previously [46,47,48,49,50]. All PCR products were sequenced and analyzed using BLAST search.

### 4.5. Clonal Relationships

The genetic relationship between CPE isolates was evaluated using repetitive sequence-based PCR (rep-PCR), using the DiversiLab™ strain typing system (bioMérieux, Marcy l’Etoile, France). Results were interpreted with DiversiLab web-based software. This software calculated a dendrogram based on the Pearson correlation and the modified Kullback–Leibler method and created a scatter plot and a virtual gel image. The relatedness was determined by cluster analysis according to guidelines provided by the manufacturer. Isolates with identical strain patterns were considered indistinguishable if the similarity percentage was ≥ 95%. Multilocus sequence typing (MLST) analysis was also performed using the PubMLST scheme (https://pubmlst.org/, accessed on 20 January 2022).

### 4.6. Phylogenetic Grouping and Screening for Virulence Determinants

For carbapenemase-producing *K. pneumoniae* isolates, multiplex PCR was used to detect VF-associated genes including: (i) *fimbriae* and/or adhesins: *fimH* (type 1 *fimbriae*), *mrkD* (adhesion type 3 *fimbriae*), and *cf29a* (adhesion CF29K); (ii): toxins: *ureA* (urease synthesis); (iii) iron uptake: *entB* (siderophore), *ybtS* (yersiniabactin), *kfu* (iron transport and phosphotransferase function), and *iutA* (hydroxamate siderophore); (iv) protectins: *magA* (capsular serotype K1 and hypermucoviscosity phenotype), *rmpA* (regulator of mucoid phenotype A), *kpsMII* (group2 capsule), and the genes of capsular serotypes K5, K57, K54, and K20; and (v) others: *allS* (associated with allantoin metabolism), *uge* (uridine diphosphate galacturonate 4-epimerase), and *wabG* (core LPS biosynthesis) [51,52].

All carbapenemase-producing *E. coli* isolates were assigned to the phylogenetic groups (A, B1, B2, C, D, E, and F) by using multiplex PCR based on the method described by Clermont et al. [53]. *E. coli* isolates were also examined for the presence of genes encoding certain putative virulence determinants including *papAH*, *papC*, *papEF*, *papG* alleles I, II, III (encoding P *fimbriae*), *sfaS/focG* (S *fimbriae* and FIC *fimbrae*), *afa/draBC* (Dr family adhesin), *fimH* (mannose-specific adhesin subunit of type 1 *fimbriae*), *hlyA* (hemolysin), *cnf1* (cytotoxic necrotizing factor-1), *iutA* (aerobactin siderophore receptor), *kpsMTII* and *kpsMKI* (capsule synthesis), *traT* (serum resistance-associated factor), *malX* (pathogenicity island marker from strain CFT073), *iroN* (iron siderophore receptor), and *irp2* (yersiniabactin). ExPEC status (i.e., exhibition of ≥ 2 VF genes among the following VFs: *pap*, *sfa/focDE*, *afa/draBC*, *iutA*, and *kpsMT2*) was determined as previously described [54].

### 4.7. Plasmid Typing

Plasmid incompatibility groups were determined using Plasmid Relaxase gene Typing (PRaseT) [55]. This latter includes six different multiplex PCRs using primers targeting the most frequently encountered Relaxase genes in Enterobacterales. A simplex PCR was performed to target IncR [56].

## 5. Conclusions

In conclusion, this study highlighted the emergence of pan-drug-resistant Enterobacterales in Annaba University Hospital mainly mediated by *bla*_OXA-48_ and *bla*_NDM-1_ genes. It also described the first report of *E. coli*-producing *bla*_NDM-1_ genes in community-acquired infection in Algeria. These results corroborate the importance of the commitment of a multidisciplinary team and raise the urgency of the implementation of a nationwide surveillance program combating nosocomial and community infections caused by such multidrug-resistant bacteria and limiting their spread.

## Figures and Tables

**Figure 1 antibiotics-11-00750-f001:**
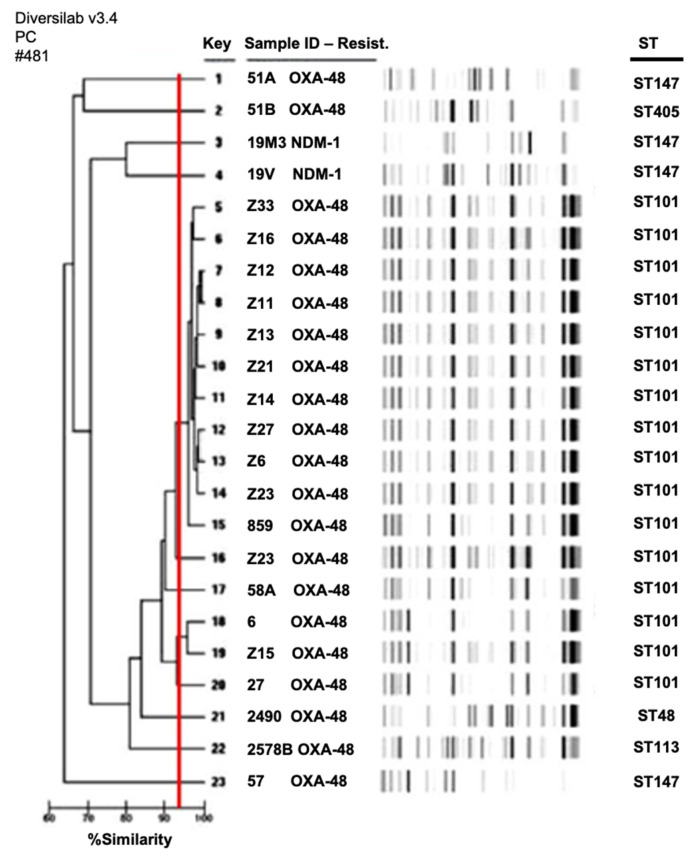
Dendrogram of NDM-1- and OXA-48-producing *Klebsiella pneumoniae* isolates. Red line corresponds to the limit of the clonality recommended by the manufacturer.

**Figure 2 antibiotics-11-00750-f002:**
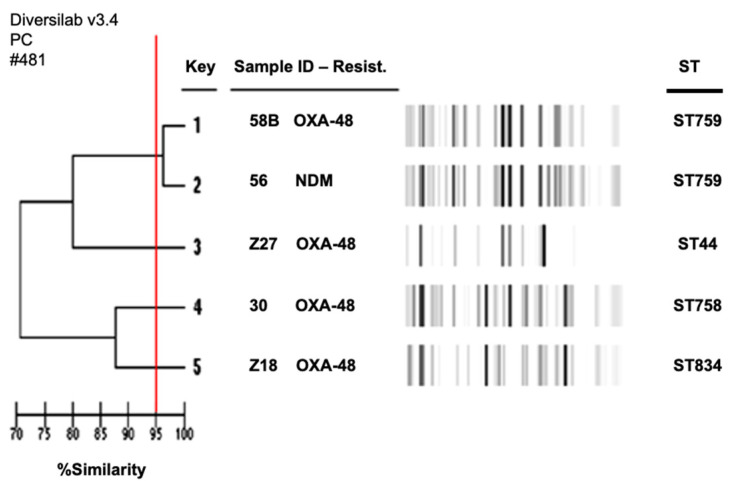
Dendrogram of NDM-1- and OXA-48-producing *Escherichia coli* isolates. Red line corresponds to the limit of the clonality recommended by the manufacturer.

**Table 1 antibiotics-11-00750-t001:** Clinical features of carbapenemase-producing isolates obtained in this study.

Isolates	Species	Carbapenemase Genes	Additional Antibiotic Resistance Genes	Sequence Type	Virulence Genes	Phylogenetic Groups (*E. coli*)	Plasmid Type
2490	*K. pneumoniae*	*bla* _OXA-48_	*qnrB*, *qnrS*, *oqxAB*	ST48	*ybtS*, *mrkD*, *entB*, *ugeF*, *wabG*, *ureA*, *fimH*	NA *	IncM, IncR, MobC11
859	*K. pneumoniae*	*bla* _OXA-48_	*bla*_CTX-M-15_, *qnrB*, *oqxAB*, *aac(6′)*-Ib-*cr*, *aac(3)*-II	ST101	*ybtS*, *mrkD*, *entB*, *ugeF*, *wabG*, *ureA*, *fimH*, *kfu*	NA	*IncM*, *MobQu*
2578B	*K. pneumoniae*	*bla* _OXA-48_	*qnrB*, *oqxAB*	ST113	*mrkD*, *entB*, *ugeF*, *wabG*, *ureA*, *fimH*	NA	*IncM*
19M3	*K. pneumoniae*	*bla* _NDM-1_	*qnrB*, *aac(6′)*-Ib-*cr*, *oqxAB*, *aac(3)*-II	ST147	*ybtS*, *mrkD*, *entB*, *ugeF*, *wabG*, *ureA*, *fimH*	NA	*IncR*
19V	*K. pneumoniae*	*bla* _NDM-1_	*bla*_CTX-M-15_, *qnrB*, *oqxAB**aac(6′)*-Ib-*cr*, *aac(3)*-II	ST147	*ybtS*, *mrkD*, *entB*, *ugeF*, *wabG*, *ureA*, *fimH*	NA	*IncR*
27	*K. pneumoniae*	*bla* _OXA-48_	*bla*_CTX-M-15_, *oqxAB*, *aac(6′)*-Ib-*cr*, *aac(3)*-II	ST101	*ybtS*, *mrkD*, *entB*, *kfu*, *ugeF*, *wabG*, *ureA*, *fimH*	NA	*IncM*, *colE NT2*
57	*K. pneumoniae*	*bla* _OXA-48_	*bla*_CTX-M-15_, *oqxAB*, *aac(6′)*-Ib-*cr*, *aac(3)*-II	ST147	*ybtS*, *mrkD*, *entB*, *kfu*, *ugeF*, *wabG*, *ureA*, *fimH*	NA	*IncM*, *IncR FIIK*
51A	*K. pneumoniae*	*bla* _OXA-48_	*bla*_CTX-M-15_, *oqxAB*, *aac(6′)*-Ib-*cr*, *rmtB*	ST147	*ybtS*, *mrkD*, *entB*, *iutA*, *ugeF*, *wabG*, *ureA*, *fimH*	NA	*IncM*, *FIIK*, *FV*, *IncAC*, *X3*
51B	*K. pneumoniae*	*bla* _OXA-48_	*bla*_CTX-M-15_, *qnrB*, *oqxAB*, *aac(6′)*-Ib-*cr*, *aac(3)*-II	ST405	*ybtS*, *mrkD*, *entB*, *ugeF*, *ureA*, *fimH*	NA	*IncM*, *FIIK*
58A	*K. pneumoniae*	*bla* _OXA-48_	*bla*_CTX-M-15_, *oqxAB*, *aac(3)*-II	ST101	*ybtS*, *mrkD*, *entB*, *kfu*, *ugeF*, *wabG*, *ureA*, *fimH*	NA	*IncM*, *colE NT1*, *colE NT2*
6	*K. pneumoniae*	*bla* _OXA-48_	*bla*_CTX-M-15_, *oqxAB*, *aac(6′)*-Ib-*cr*, *aac(3)*-II	ST101	*ybtS*, *mrkD*, *entB*, *kfu*, *ugeF*, *ureA*, *fimH*	NA	*IncM*, *IncR*, *colE NT2*
Z6	*K. pneumoniae*	*bla* _OXA-48_	*qnrS*, *oqxAB*	ST101	*ybtS*, *mrkD*, *entB*, *kfu*, *ugeF*, *wabG*, *ureA*, *fimH*	NA	*IncM*, *FIIK*, *colE NT2*
Z7	*K. pneumoniae*	*bla* _OXA-48_	*bla*_CTX-M-15_, *qnrA*, *qnrB*, *qnrS*, *oqxAB*, *aac(6′)*-Ib-*cr*, *aac(3)*-II, *ant(2″)*	ST101	*ybtS*, *mrkD*, *entB*, *kfu*, *ugeF*, *wabG*, *ureA*, *fimH*	NA	*IncM*, *FIIK*, *colE NT2*
Z9	*K. pneumoniae*	*bla* _OXA-48_	*bla*_CTX-M-15_, *qnrA*, *qnrB*, *qnrS*, *oqxAB*, *aac(6′)*-Ib-*cr*, *aac(3)*-II	ST101	*ybtS*, *mrkD*, *entB*, *kfu*, *ugeF*, *wabG*, *ureA*, *fimH*	NA	*IncM*, *FIIK*, *colE NT2*
Z11	*K. pneumoniae*	*bla* _OXA-48_	*bla*_CTX-M-15_, *qnrA*, *qnrB*, *qnrS*, *oqxAB*, *aac(6′)*-Ib-*cr*, *aac(3)*-II	ST101	*ybtS*, *mrkD*, *entB*, *kfu*, *ugeF*, *wabG*, *ureA*, *fimH*	NA	*IncM*, *FIIK*, *colE NT2*
Z12	*K. pneumoniae*	*bla* _OXA-48_	*bla*_CTX-M-15_, *qnrA*, *qnrB*, *qnrS*, *oqxAB*, *aac(6′)*-Ib-*cr*	ST101	*ybtS*, *mrkD*, *entB*, *kfu*, *ugeF*, *wabG*, *ureA*, *fimH*	NA	*IncM*, *FIIK*, *colE NT2*
Z13	*K. pneumoniae*	*bla* _OXA-48_	*bla*_CTX-M-15_, *qnrA*, *qnrB*, *qnrS*, *oqxAB*, *aac(6′)*-Ib-*cr*, *aac(3)*-II	ST101	*ybtS*, *mrkD*, *entB*, *kfu*, *ugeF*, *wabG*, *ureA*, *fimH*	NA	*IncM*, *FIIK*, *colE NT2*
Z14	*K. pneumoniae*	*bla* _OXA-48_	*bla*_CTX-M-15_, *qnrA*, *qnrB*, *qnrS*, *oqxAB*, *aac(6′)*-Ib-*cr*, *aac(3)*-II	ST101	*ybtS*, *mrkD*, *entB*, *kfu*, *ugeF*, *wabG*, *ureA*, *fimH*	NA	*IncM*, *FIIK*, *colE NT2*
Z15	*K. pneumoniae*	*bla* _OXA-48_	*bla*_CTX-M-15_, *qnrA*, *qnrB*, *qnrS*, *oqxAB*, *aac(6′)*-Ib-*cr*, *aac(3)*-II	ST101	*ybtS*, *mrkD*, *entB*, *kfu*, *ugeF*, *wabG*, *ureA*, *fimH*	NA	*IncM*, *FIIK*, *colE NT2*
Z16	*K. pneumoniae*	*bla* _OXA-48_	*bla*_CTX-M-15_, *qnrA*, *qnrB*, *qnrS*, *oqxAB*, *aac(6′)*-Ib-*cr*, *aac(3)*-II	ST101	*ybtS*, *mrkD*, *entB*, *kfu*, *ugeF*, *wabG*, *ureA*, *fimH*	NA	*IncM*, *FIIK*, *colE NT2*
Z21	*K. pneumoniae*	*bla* _OXA-48_	*bla*_CTX-M-15_, *qnrA*, *qnrS*, *oqxAB*, *aac(6′)*-Ib-*cr*	ST101	*ybtS*, *mrkD*, *entB*, *kfu*, *ugeF*, *wabG*, *ureA*, *fimH*	NA	*IncM*, *FIIK*, *MobQu*
Z23	*K. pneumoniae*	*bla* _OXA-48_	*bla*_CTX-M-15_, *oqxAB*, *aac(6′)*-Ib-*cr*, *aac(3)*-II	ST101	*ybtS*, *mrkD*, *entB*, *kfu*, *ugeF*, *wabG*, *ureA*, *fimH*	NA	*IncM*, *MobQu*
Z33	*K. pneumoniae*	*bla* _OXA-48_	*qnrA*, *qnrS*, *oqxAB*, *aac(3)*-II	ST101	*ybtS*, *mrkD*, *entB*, *kfu*, *ugeF*, *wabG*, *ureA*, *fimH*	NA	*IncM*, *FIIK*, *MobQu*
2578A	*E. cloacae*	*bla* _OXA-48_	*bla*_CTX-M-9_, *qnrB*	ST68	NA	NA	*IncM*, *colE NT2*
30	*E. coli*	*bla* _OXA-48_	*aac(6′)*-Ib-*cr*	ST758	*traT*, *fimH*	B1	*IncM*, *AC*, *FV*, *colE NT1*
58B	*E. coli*	*bla* _OXA-48_	*bla*_CTX-M-15_, *aac(6′)*-Ib-*cr*	ST759	*traT*, *fimH*, *iutA*	E	*IncM*, *colE NT1*, *colE NT2*, *IncAC*, *FV*, *X3*
56	*E. coli*	*bla* _NDM-1_	*bla*_CTX-M-15_, *aac(6′)*-Ib-*cr*, *rmtB*	ST759	*traT*, *fimH*, *hlyA*	E	*colE NT2*, *FV*, *X3*
Z18	*E. coli*	*bla* _OXA-48_	*aac(3)*-II	ST834	*traT*, *fimH*, *hlyA*	A	*IncM*, *FV*, *X3*
Z27	*E. coli*	*bla* _OXA-48_	*bla*_CTX-M-15_, *aac(6′)*-Ib-*cr*, *aac(3)*-II	ST44	*traT*, *fimH*, *hlyA*, *irp2*, *malX*, *papAH*, *papC*, *iutA*, *papG2*, *kpsMT2*	E	*IncM*, *FV*

* NA, not applicable.

## Data Availability

Not applicable.

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
