# Peer review of "Dissemination of OXA-48- and NDM-1-Producing Enterobacterales Isolates in an Algerian Hospital"

_antibiotics, 2022, doi:10.3390/antibiotics11060750_

Round 1

Reviewer 1 Report

Manuscript is describing situation with carbapenemase producing bacteria in Algerian hospital. Manuscript is very well written and highlighten original findings of authors (NDM-1 producing E.coli ina Algeria).  I recommend to accept manuscript as a publication in Antibiotics Journal.

Author Response

Manuscript is describing situation with carbapenemase producing bacteria in Algerian hospital. Manuscript is very well written and highlighten original findings of authors (NDM-1 producing E.coli ina Algeria).  I recommend to accept manuscript as a publication in Antibiotics Journal.

We thank the reviewer for these positive comments.

Reviewer 2 Report

Introduction:

Introduction should be shortened. I.e. lines 61-70 include a very detailed explanation of aminoglycoside and quinolone resistance which is not exactly the focus of this paper. Instead, since the manuscript has regional importance, the current situation in Algeria regarding spread of CPE could be described. 

Results:

Line 79: Why all the other isolates (non-NDM, non-OXA-48) had reduced susceptibility to carbapenems? What was the mechanism of resistance? This needs to be clarified. 

Line 83: This sentence does not add to the results since CPE isolates are usually resistant to these agents. 

Line 87: What are the MIC values of colistin and carbapenems? Please add this to the results.

Methods:

MICs should have been determined for all tested antibiotics as it is a better method for AST. The MIC values should be added to the results section. 

Line 215: How is reduced susceptibility to carbapenems defined?

Line 220: Which breakpoints were used? EUCAST, version, year?

Line 234: Whole-genome sequence is more and more used nowadays to characterize isolates genotypically. Given the small number of isolates in this study, this should have been done. This would replace all methods used under 4.4, 4.5, 4.6 and 4.7 and provide even more in-depth information about the molecular background of the isolates. 

Author Response

Introduction:

Introduction should be shortened. I.e. lines 61-70 include a very detailed explanation of aminoglycoside and quinolone resistance which is not exactly the focus of this paper. Instead, since the manuscript has regional importance, the current situation in Algeria regarding spread of CPE could be described. 

We modified the Introduction accordingly.

Results:

Line 79: Why all the other isolates (non-NDM, non-OXA-48) had reduced susceptibility to carbapenems? What was the mechanism of resistance? This needs to be clarified. 

We clarified this part. These isolates presented an impermeability to carbapenems.

Line 83: This sentence does not add to the results since CPE isolates are usually resistant to these agents. 

We deleted this sentence in the new version of the manuscript.

Line 87: What are the MIC values of colistin and carbapenems? Please add this to the results.

We added a new Table S2 with these results.

Methods:

MICs should have been determined for all tested antibiotics as it is a better method for AST. The MIC values should be added to the results section. 

MICs were determined for the main antibiotics concerned by the subject of our paper. The results are presented in the new Table S2. For the other antibiotics, the disk diffusion method is validated as an adapted method to determine antibiotic resistance by EUCAST.

Line 215: How is reduced susceptibility to carbapenems defined?

We defined reduced susceptibility to carbapenems in the new version of the manuscript in 4.1 section.

Line 220: Which breakpoints were used? EUCAST, version, year?

We added this information in the Methods section (4.2).

Line 234: Whole-genome sequence is more and more used nowadays to characterize isolates genotypically. Given the small number of isolates in this study, this should have been done. This would replace all methods used under 4.4, 4.5, 4.6 and 4.7 and provide even more in-depth information about the molecular background of the isolates. 

We agree with the reviewer. However, this technology are not available in the Algerian Lab of our collaborators. In this paper, all the experiments were conducted in Algeria. The French lab provided all the protocols, and trained the bacteriologist to perform molecular biology. Thus, the different results presented here allowed to characterize the isolates, their clonality and their main resistant and virulence content usually determined in the epidemiological studies.

Reviewer 3 Report

In the current study, the authors described the dissemination carbpenemase-producing Enterobacerales in an Algerian hospital. The study is well organized and the findings are of interest.

Comments:

Line 30: correct to IncM.

Line 83: correct to ‘Variable’.

Line 92: correct to ‘remaining isolates’.

Line 96: correct to ‘enzyme-encoding genes’.

Line 100: oqxAB gene is not the chromosomal gene found in all K. pneumoniae isolates. Please correct this.

Line 104: How the evaluation was performed?

Lines 133-134: Based on literature (PMID: 27855076) this plasmid belongs to IncM group. Please correct this point.

Line 135-136: But you have not proven the association of blaNDM-1 gene with IncR and IncF replicons. Please correct this part?

Figures 1 and 2: how the dendrograms were constructed?

Lines 175-177: similar results have also been reported in the Czech Republic (PMID: 30042758). Please mention this point.

Lines 209-212: what was the localization of the virulence genes?

Author Response

Line 30: correct to IncM.

We corrected along the text.

Line 83: correct to ‘Variable’.

We modified accordingly.

Line 92: correct to ‘remaining isolates’.

We modified accordingly.

Line 96: correct to ‘enzyme-encoding genes’.

We modified accordingly.

Line 100: oqxAB gene is not the chromosomal gene found in all K. pneumoniae isolates. Please correct this.

We modified accordingly.

Line 104: How the evaluation was performed?

We added this information in the new version of the manuscript (section 2.3)

Lines 133-134: Based on literature (PMID: 27855076) this plasmid belongs to IncM group. Please correct this point.

We agree with the reviewer and we corrected this point along the text.

Line 135-136: But you have not proven the association of blaNDM-1 gene with IncR and IncF replicons. Please correct this part?

We agree with the reviewer. We rewrote the sentence (section 2.5).

Figures 1 and 2: how the dendrograms were constructed?

We added this information in the Materials and Methods’ section (section 4.5).

Lines 175-177: similar results have also been reported in the Czech Republic (PMID: 30042758). Please mention this point.

We added this reference (Ref 27 in the new version).

Lines 209-212: what was the localization of the virulence genes?

The virulence genes are localized on E. coli chromosome. We modified the sentence at the end of the Discussion section in the new version of the manuscript.

Round 2

Reviewer 2 Report

No further comments. Thank you

Author Response

We thank the reviewer for this appreciation.